# Cyclo(His-Pro) Exerts Protective Carbonyl Quenching Effects through Its Open Histidine Containing Dipeptides

**DOI:** 10.3390/nu14091775

**Published:** 2022-04-23

**Authors:** Luca Regazzoni, Laura Fumagalli, Angelica Artasensi, Silvia Gervasoni, Ettore Gilardoni, Angelica Mazzolari, Giancarlo Aldini, Giulio Vistoli

**Affiliations:** 1Dipartimento di Scienze Farmaceutiche, Università degli Studi di Milano, I-20133 Milano, Italy; luca.regazzoni@unimi.it (L.R.); laura.fumagalli@unimi.it (L.F.); angelica.artasensi@unimi.it (A.A.); silvia.gervasoni@unimi.it (S.G.); ettore.gilrdoni@gmail.com (E.G.); angelica.mazzolari@unimi.it (A.M.); giancarlo.aldini@unimi.it (G.A.); 2Department of Physics, Università di Cagliari, Citt. Universitaria, I-09042 Monserrato, Italy

**Keywords:** Cyclo(His-Pro), histidine containing peptides, protein-rich foods, HNE, carbonyl quenching, serum carnosinase, organocatalysis

## Abstract

Cyclo(His-Pro) (CHP) is a cyclic dipeptide which is endowed with favorable pharmacokinetic properties combined with a variety of biological activities. CHP is found in a number of protein-rich foods and dietary supplements. While being stable at physiological pH, CHP can open yielding two symmetric dipeptides (His-Pro, Pro-His), the formation of which might be particularly relevant from dietary CHP due to the gastric acidic environment. The antioxidant and protective CHP properties were repeatedly reported although the non-enzymatic mechanisms were scantly investigated. The CHP detoxifying activity towards α,β unsaturated carbonyls was never investigated in detail, although its open dipeptides might be effective as already observed for histidine containing dipeptides. Hence, this study investigated the scavenging properties of TRH, CHP and its open derivatives towards 4-hydroxy-2-nonenal. The obtained results revealed that Pro-His possesses a marked activity and is more reactive than l-carnosine. As investigated by DFT calculations, the enhanced reactivity can be ascribed to the greater electrophilicity of the involved iminium intermediate. These findings emphasize that the primary amine (as seen in l-carnosine) can be replaced by secondary amines with beneficial effects on the quenching mechanisms. Serum stability of the tested peptides was also evaluated, showing that Pro-His is characterized by a greater stability than l-carnosine. Docking simulations suggested that its hydrolysis can be catalyzed by serum carnosinase. Altogether, the reported results evidence that the antioxidant CHP properties can be also due to the detoxifying activity of its open dipeptides, which might be thus responsible for the beneficial effects induced by CHP containing food.

## 1. Introduction

Cyclo(His-Pro) (CHP) is a ubiquitous dipeptide which is present in many foods and supplements [1,2] where its concentration ranges from ng/g to μg/g, as recently reviewed [3]. CHP is abundant in some fish and fish-derived products [1], in ethyl alcohol-refluxed soy protein hydrolysate (SPH) [4] and in several yeast extracts [5,6], usually employed in the food and fermentation industries [7]. More generally, CHP is highly present in all processed protein-rich foods [8] and in dietary supplements derived from casein and/or soy protein hydrolysates [9]. Endogenously, CHP can either be produced de novo or can derive from the thyrotropin-releasing hormone (TRH) catabolism following the cyclization of the His-Pro dipeptide, which, in turn, arises from the hydrolytic activity by pyroglutamyl aminopeptidase (see Figure 1) [10].

The cyclization occurs spontaneously at physiological conditions and has two major consequences [11]. The cyclized dipeptide is more resistant to the hydrolytic metabolism and is recognized by several transporters, which modulate its pharmacokinetic profile by promoting gastro-intestinal absorption and blood–brain barrier (BBB) permeation [12]. Thus, CHP is orally bioavailable and reaches the CNS, with a loco-regional disposal which parallels that of the organic cationic transporter OCT2 [13].

CHP shows a wide range of biological activities, which include, among others, anti-inflammation effects by modulating the Nrf2 and Nf-Kb signaling [14] and hypoglycemic activities [15] by influencing the expression of various proteins involved in glucose homeostasis such as apolipoproteins and fibrinogen [16]. These biological effects render CHP a potential therapeutic agent in neuro-degenerative diseases [17] as well as in diabetes [6]. Despite that some biological pathways have been clarified, the precise mechanisms are still debated. Some CHP central effects (such as thermoregulation and nociception) might be ascribed to its interaction with different targets including GABA receptors and neurotransmitter transporters [18,19].

Furthermore, CHP is described as an antioxidant peptide and the proposed general mechanism involves the activation of the Nrf2-mediated up-regulation of antioxidant defense [20]. Nevertheless, the CHP antioxidant effects can also by ascribed to three major non-enzymatic mechanisms which are not yet fully investigated. In detail, radical scavenging and metal chelation reduce the oxidative reactions, while carbonyl quenching detoxifies the resulting reactive carbonyl species (RCS) [21]. The CHP radical scavenging activity was evaluated only by Jung and co-workers who revealed that yeast hydrolysates with high CHP content possess a remarkable scavenging activity towards well-known radical inducers [5]. Although the presence of the imidazole ring suggests that CHP can chelate metals, the biological role of its metal chelation was only indirectly investigated. Some studies described the CHP capacity to increase the zinc absorption with beneficial effects in glucose homeostasis and hippocampal neurogenesis [22].

Lastly, the CHP carbonyl quenching capacity was never investigated, although the properties of the histidine containing peptides [23,24], in particular, the scavenging behavior, were analyzed in-depth as exemplified by l-carnosine and derivatives [25,26]. This absence might be explained by considering that CHP does not possess free amino groups, a feature required by the quenching mechanism. While being rather stable in physiological conditions [27], CHP can yet open yielding two symmetric dipeptides [28,29]: His-Pro and Pro-His (see Figure 1). Notably, the open dipeptides can be principally produced from dietary CHP due to the hydrolytic effect of the gastric environment as confirmed by a recent study which revealed the positive effect of potassium ions in the opening of cyclic peptides [25]. Thus, cyclization can be seen as a recycling strategy by which two different peptides are generated from only one. Such a process appears to be particularly interesting since CHP can derive, in turn, from the TRH recycling.

Considering the well-known pathogenic role of α,β unsaturated carbonyls [11], the study investigated the quenching activity towards 4-hydroxy-2-nonenal (HNE) of TRH, CHP and the two open dipeptides (His-Pro and Pro-His). The quenching activity towards dicarbonyl species was not evaluated since a previous study revealed that histidine containing dipeptides possess poor quenching activities towards these reactive species [30]. The amide of the more promising derivative (Pro-His-NH_2_, see below), which can derive from the alternative nucleophilic opening of the diketopiperazine ring [27] was also investigated. The serum stability of all considered compounds was tested and, since cyclic dipeptides were reported as inhibitors of human serum carnosinase (hCN1) [31], the CHP inhibition effect was also evaluated.

## 2. Materials and Methods

### 2.1. Reagents and Chemicals

Water, HPLC grade (18 MΩ), was purified with a Milli-Q water system (Millipore, Milan, Italy). l-carnosine was kindly provided by Flamma s.p.a. (Chignolo D’isola, Bergamo, Italy). TRH, CHP, His-Pro, solvents at HPLC grade and all other chemicals were purchased from Merck KGaA, Darmstadt, Germany. ^1^H-NMR spectra were recorded using a FT-spectrometer operating at 300 MHz while ^13^C-NMR at 75.43 MHz. Chemical shifts are reported in ppm relative to the residual solvent (CHCl_3_, MeOH, or DMSO) as an internal standard. Signal multiplicity is designed according to the following abbreviations: s = singlet, d = doublet, dd = doublet of doublets, t = triplet, m = multiplet, br s = broad singlet and br t = broad triplet. Purifications were performed by flash chromatography using silica gel (particle size 40–63 μm, Merck) on Isolera^TM^ (Biotage, Uppsala, Sweden) apparatus.

### 2.2. Synthesis of L-Pro-l-His and l-Pro-l-His-NH_2_

The general scheme of the procedures for the synthesis of Pro-His and Pro-His-NH_2_ is reported in Figure 1.

#### 2.2.1. N-Boc-l-Pro-OH (**2**)

Boc_2_O (1.13 g, 10.42 mmol) previously dissolved in tetrahydrofuran (10 mL) was added dropwise to a solution of (S)-proline **1** (1.2 g, 10.42 mmol) in aq. NaOH (1M, 20 mL) and tetrahydrofuran (5 mL) at 0 °C. The resulting mixture was stirred at 0 °C for 30 min, then overnight at room temperature. The organic solvent was removed under reduced pressure. The remaining aqueous solution was acidified to pH ≅ 2 with aq. KHSO_4_ (1 M). The aqueous solution was extracted with dichloromethane (3 × 20 mL). The combined organic layers were washed with brine (30 mL), dried over Na_2_SO_4_ and concentrated under vacuum to afford carbamate **2** (2.22 g, 99% yield) as a white solid (mp 134.0 °C), which was sufficiently pure to be taken on to the next step.

^1^H-NMR: conform to structure as reported in literature [32].

#### 2.2.2. N-Boc-l-Pro-l-His-OMe (**3**)

To a solution of **2** (200.00 mg, 0.93 mmol) in DMF (3.0 mL) at 0 °C were added diisopropylethylamine (0.18 mL, 1.02 mmol) and TBTU (328.48 mg, 1.02 mmol). After stirring for 20 min at 0 °C, a solution of l-histidine methylester dihydrochloride (225.15 mg, 0.93 mmol) and diisopropylethylamine (0.32 mL, 1.86 mmol) in DMF (3 mL) was added. The resulting reaction mixture was stirred overnight at room temperature. The solvent was removed under vacuum and the residue was taken up by dichloromethane (10 mL). Afterwards the organic phase was washed sequentially with aq. NaHCO_3_ 10% (5 mL) and brine (5 mL). The organic layer was dried over Na_2_SO_4_ and concentrated under vacuum to afford a pale oil. Finally, the crude product was subjected to purification by chromatography on silica gel (DCM/MeOH = 95:5) to give 136.30 mg (0.37 mmol, 40%) of **3** as colorless oil. ^1^H-NMR (300 MHz, CDCl_3_) δ 7.79 (s, 1H), 6.87 (s, 1H), 4.88–4.75 (m, 1H), 4.22 (m, 1H), 3.77 (s, 3H), 3.50 (m, 2H), 3.36 (dd, *J* = 15.1, 4.6 Hz, 1H), 3.20 (dd, *J* = 15.0, 3.3 Hz, 1H), 2.24–2.00 (m, 3H), 1.90 (m, 1H), 1.46 (s, 9H).

#### 2.2.3. l-Pro-l-His-OH dihydrochloride (**4**)

N-Boc-l-Pro-l-His-OMe (120.00 mg, 0.33 mmol) was dissolved in 2 mL of Aq. 2N HCl and the resulting reaction mixture was stirred at 50 °C for 2 h. Afterwards, azeotropic distillation with chloroform allowed to obtain the l-Pro-l-His-OH dihydrochloride (**4**) as a waxy white solid 80 mg (0.25 mmol, 76%). ^1^H-NMR (300 MHz, *d*_6_-DMSO) δ 10.2 (brs, 1H, exchangeable with D_2_O), 9.25 (d, *J* = 7.9 Hz, 1H, exchangeable with D_2_O), 9.04 (d, *J* = 1.8 Hz, 1H), 8.44 (brs exchangeable with D_2_O, 1H), 7.47 (d, *J* = 1.8 Hz, 1H), 4.60–4.55 (m, 1H), 4.25–4.18 (m, 1H) 3.21 (m, 4H), 3.09 (dd, *J* = 14.9, 9.2 Hz, 1H), 2.85–1.68 (m, 3H). ^13^C NMR (75 MHz, *d*_6_-DMSO) δ 171.72, 168.82, 133.97, 129.42, 117.47, 58.99, 52.29, 45.89, 29.94, 25.99, 23.81.

#### 2.2.4. l-Pro-l-His-NH_2_ (**5**)

NH_3_ gas was bubbled into a solution of N-Boc-l-Pro-l-His-OMe (120.00 mg, 0.33 mmol) in methanol (5 mL), once a day for three days, until ^1^H-NMR indicated the complete conversion of the methyl ester. After removal of the solvent under vacuum 2N HCl (2 mL) was added and the resulting reaction mixture was stirred at 50 °C for 2 h. Afterwards, azeotropic distillation with chloroform allowed to obtain a pale-yellow oil that was purified by preparative TLC (isopropanol/chloroform/ammonia; 7/3/2%) to afford l-Pro-l-His-NH_2_ as a colorless oil 30 mg (0.12 mmol, 36%). ^1^H NMR (300 MHz, CD_3_OD) δ 8.84 (s, 1H), 7.44 (s, 1H), 4.76 (dd, *J* = 8.0, 5.5 Hz, 1H), 4.37 (dd, *J* = 8.0, 6.3 Hz, 1H), 3.46–3.30 (m, 2H), 3.26–3.11 (m, 1H), 2.55–2.38 (m, 1H), 2.15–1.97 (m, 3H). ^13^C NMR (75 MHz, CD_3_OD) δ 168.40, 168.40, 133.56, 133.56, 129.33, 129.33, 117.24, 117.24, 59.72, 59.72, 52.40, 52.40, 51.74, 48.46, 48.18, 47.89, 47.61, 47.32, 47.04, 46.76, 46.00, 46.00, 29.48, 29.48, 26.73, 26.73, 23.60, 23.60, 23.51, 23.51.

### 2.3. Quenching Activity towards HNE

The reactivity towards HNE was tested for each peptide separately, by monitoring within 3 h from the incubation the residual HNE via HPLC-UV, as previously reported [15]. Minor modifications of the method were implemented. A Kinetex C18 column (25 mm × 2.10 mm, particle size 2.6 µm, pore size 100 Å, Phenomenex, Castel Maggiore, Italy) was used instead and the elution of residual HNE was provided at 40 °C by 300 µL/min flow rate of aqueous ammonium acetate (10 mM) containing 20% acetonitrile. For active compounds, the formation of HNE adduct was confirmed by mass spectrometry. Samples were diluted in ACN/H_2_O/formic acid, 50/50/0.1, *v*/*v*/*v* solution down to a suitable concentration and pumped at 5 µL/min directly into ESI capillary (Finnigan Ion Max, Thermo Fisher Scientific, Rodano, Italy) and 3.5 kV voltage, 270 °C capillary temperature, 35 V capillary voltage and 50 V tube lens were applied to ensure sample nebulization. Spectra were acquired in positive ion mode for 5 × 10^5^ ions per scan, by using a LTQ-Orbitrap XL-ETD analyzer (Thermo Fisher Scientific, Rodano, Italy) operating in a 150–700 m/z scan range at a resolution of (FWHM at m/z 400).

### 2.4. Serum Stability

Serum stability was measured as previously described [33]. Briefly, analytes were spiked separately in pre-heated (37 °C) serum aliquots down to a final concentration of 5 µM. The residual concentration of analyte was measured at the beginning and after 5, 15, 30, 60 and 120 min by means of LC-MS. Prior to analyses, hydrolysis was blocked at the desired time points diluting with 9 volumes of acetonitrile at 4 °C to provide protein precipitation. The analyte residual concentration was then measured in the supernatants obtained upon centrifugation. The mass spectrometer detector worked in MRM (multiple reaction monitoring) mode following the transition reported in Table 1. The same method was also used to evaluate the inhibition effect of CHP by incubating serum samples with equimolar amounts of carnosine and CHP.

### 2.5. Computational Studies

DFT calculations were performed as previously detailed [15] to monitor the quenching mechanisms of Pro-His and Pro-His-NH_2_. Briefly, the iminium ions, the macrocyclic intermediates and the final adducts in their open form were generated using the VEGA software which explored their conformational space by Monte Carlo analyses [34]. The resulting best conformation was further optimized by PM7 semi-empirical calculations [35] and underwent DFT analysis using the B3LYP method and the 6–31G basis set as implemented by the GAMESS/Firefly software [36]. To evaluate the solvent effect, the PCM implicit solvent model was applied.

To rationalize the serum stability of the tested peptides, docking simulations involving the resolved hCN1 structure (PDB code: 3DLJ) were carried out as described elsewhere [22]. Docking calculations were performed using GOLD [37] by focusing the search within a 10 Å radius sphere around the zinc ions. For each peptide, 100 poses were generated and evaluated by the ChemPLP scoring function.

## 3. Results

### 3.1. Quenching Activity towards HNE of the Tested Peptides

Table 2 compiles the scavenging activity of the tested compounds expressed as percentages of the quenched HNE at different times. Table 2 also includes the resulting HNE half-life values in presence of the quenchers. As expected, TRH and CHP, which do not possess free N-terminal groups, show very modest quenching activities. The poor but not nil activity of CHP (which should be completely inactive per se) emphasizes that a small fraction of the cyclic peptide spontaneously opens and reacts with HNE. The dipeptide His-Pro reveals a very poor activity probably due to the strong ion-pair between the charged termini which shields the nucleophilicity of the imidazole ring. In contrast, the dipeptide Pro-His shows a quenching activity significantly greater than l-carnosine, while Pro-His-NH_2_ reveals an activity comparable with l-carnosine.

These results emphasize that a primary amino group (as seen for l-carnosine) is not mandatory but can be replaced by a secondary amine with beneficial effects on the quenching mechanism. As schematized in Figure 2, the greater activity of Pro-His can be explained by the involvement of a positively charged iminium intermediate which enhances the electrophilicity of the β carbon atom and favors the resulting Michael addition. The enhancing effect can be appreciated by comparing the molecular electrostatic potentials (MEP) of the carnosine-HNE enamine intermediate with the corresponding iminium ion for Pro-His. Figure 3A shows that the region surrounding the β carbon atom of the Pro-His iminium intermediate is markedly positive, while the corresponding region in the l-carnosine imine intermediate is barely positive. The difference is also confirmed by the total atomic charge of the β = CH-group which is neutral for the l-carnosine intermediate and equal to +0.14 for the Pro-His intermediate.

Figure 3B reports the mass spectrum (positive-ion mode) of the reaction mixture (Pro-His + HNE) after 3 hrs incubation. As confirmed by the agreement between theoretical and experimental m/z values, Figure 3B reveals the presence of the unreacted dipeptide (m/z = 253.129) plus an additional peak which can be attributed to the Michael adduct (m/z = 409.245) in equilibrium between its open and hemi-acetal form. The comparison between the corresponding spectra for l-carnosine derivatives [38] reveals that the peak of the imine intermediate, which was detectable in the previous spectra, is missing in Figure 3B. This difference can be seen as a further confirmation of the greater reactivity of the iminium intermediate which rapidly yields the corresponding Michael adduct, thus preventing its identification.

The employment of proline as an organocatalyst is a well-known strategy to enhance the reactivity of α,β unsaturated carbonyls by utilizing green and sustainable conditions. [39] Hence, proline can play a similar catalytic role also in vivo since its reactivity is compatible with the physiological conditions. Nevertheless, the poor activity of the physical mixture Pro + His underlines that proline cannot act as a general catalyst in vivo. Similarly to that observed for l-carnosine, one may suppose that the Pro-His dipeptide is able to stably approach the iminium intermediate to the imidazole ring in a pose conducive to the Michael reaction [40]. Moreover, the lack of peaks in the mass spectrum at higher m/z values indicates that the proline catalytic effect terminates with the formation of the detected adduct without inducing the formation of multiple cross-linked adducts at high molecular weight.

To further investigate the quenching mechanisms of the reactive Pro-His and Pro-His-NH_2_ peptides, the possible reaction pathways were simulated by DFT calculations starting from the iminium intermediate to the final Michael adduct in its open form. In detail, both paths involving the two imidazole tautomers were simulated (see Figure 2). Since proline is often used as a stereoselective catalyst [41], both stereoisomers deriving from the Michael addition on the β carbon atom were considered.

In agreement with l-carnosine [14], Table 3 shows that path A involving the imidazole’s π tautomer is favored compared to path B for the Pro-His quenching. This difference concerns both the Michael adduct cyclization and the following hydrolysis and can be ascribed to the greater steric hindrance experienced by the macrocycles formed through path B. Such a constraining effect was already evident for l-carnosine and should be more marked here due to the rigidity of the pyrrolidine ring. Nevertheless, and probably due to the greater reactivity of the iminium intermediate, Table 3 shows that path B of Pro-His is less disfavored than path B of l-carnosine and can play a role in determining the overall reactivity of Pro-His. Thus, DFT results suggest that the greater reactivity of Pro-His can be also ascribed to a greater reactivity of its tautomer τ to yield the final adduct via path B.

While showing a comparable reactivity, Pro-His-NH_2_ differs in the initial macrocycle formation since path B appears here slightly favored compared to path A. This finding suggests that intramolecular H-bonds can play a role in determining the preferred path. The relevance of path B in the initial Michael addition influences the overall quenching mechanism and the two paths show a similar role.

Pro-His and Pro-His-NH_2_ reveal similar stereoselective profiles. Table 3 shows that the (*R*) stereoisomers are favored in both paths for the initial cyclization, while the (*S*) stereoisomers are favored for the following hydrolysis. The two contrasting effects reduce the overall stereoselectivity which disappears for path A, while the (*S*) stereoisomers are slightly favored for path B in both peptides.

### 3.2. Serum Stability of the Studied Peptides

Table 2 includes the serum stability of the considered peptides expressed as t_1/2_ values (in minutes). The reference peptide, l-carnosine, has a poor stability due to hCN1, a rather specific dipeptidase which shows a good hydrolytic activity towards various histidine containing dipeptides [22]. All the analyzed peptides show a greater stability compared to l-carnosine with CHP being completely stable in serum. In detail, His-Pro is the least stable peptide although its hydrolysis should be due to proline dipeptidases which cleave dipeptides having a proline residue in C-terminal position [42]. The TRH catabolism follows well-known metabolic pathways (see Figure 1) which can occur also in serum. Interestingly, the TRH half-life determined here is significantly higher than that obtained in vivo (6.5 min), thus suggesting that other organs (e.g., kidney) might play important catabolic roles for TRH [43]. Among the tested peptides, only the Pro-His stability might be affected to the hCN1 activity.

To better clarify the hCN1 role in determining their serum stability, docking simulations of the tested peptides were performed. As expected, TRH, CHP and His-Pro are unable to assume poses comparable to that of l-carnosine and thus conducive to the catalysis. All these peptides approach the carbonyl group to the zinc ion but fail in stabilizing the pivotal ion-pairs involving the ionized termini (complexes not shown). Figure 4A shows the putative complex for Pro-His which assumes a pose comparable to that of l-carnosine, thus suggesting that this can be hydrolyzed by hCN1. In detail, Pro-His approaches the carbonyl oxygen atom to the zinc ion and stabilizes the key ion pars involving the charged termini. In addition, the imidazole ring is inserted within the proper subpocket where it stabilizes H-bonds plus hydrophobic contacts. The comparison of the pose of Pro-His with that of l-carnosine (Figure 4B) shows that both substrates similarly accommodate the peptide group in a pose conducive the catalysis and the ammonium head close to Asp136 and Asp199. In contrast, the Pro-His C-terminus assumes a slightly distorted arrangement probably due to the constraints induced by the pyrrolidine ring which hampers the ion-pair with Arg347. The weakening of this salt bridge can explain why Pro-His is more stable than l-carnosine and why Pro-His-NH_2_, which cannot elicit this ionic interaction, is more stable in serum. Finally, CHP induced no statistically significant inhibition of hCN1 activity since serum samples incubated with carnosine and CHP retained 85 ± 5% of carnosine hydrolytic activity. The decrease was not statistically significant compared to control samples showing an activity of 1.2 nmol * µL^−1^ * h^−1^ ± 13%.

## 4. Discussion

The beneficial effect of CHP containing food in diabetes was experimentally confirmed by at least three independent studies which showed that consumption of raw vegetables [4] and yeast hydrolysates [5,15] improves insulin sensitivity and glucose tolerance. Until now, these effects were primarily explained by considering two possible mechanisms: (a) a direct mechanism by which CHP acts on pancreatic inslets and hepatic cells by increasing insulin secretion and reducing plasma glucose through an incretin-like mechanism [44]; (b) an indirect mechanism by which CHP promotes the intestinal zinc absorption which, in turn, reduces blood glucose by stimulating its uptake in muscle cells [45].

The here reported results suggest that CHP can exert its antidiabetic effect through a third (indirect) mechanism by which its open dipeptides exert a marked detoxifying effect towards reactive carbonyl species (RCS), thus reducing the generation advanced lipoxidation end-products (ALEs). This indirect mechanism appears to be particularly relevant when considering that accumulating evidence underlines the relations between oxidative stress and diabetes since sustained hyperglycemia leads to an increase of the generation of radical species (ROS) which catalyze the RCS production and overall contribute to β-cell dysfunctions [46]. Hence, the RCS scavenging by CHP open dipeptides can counteract the damaging effects of the induced oxidative stress with beneficial effects for both diabetes and related complications such as nephropathy and vasculopathy [47].

Although CHP can also open at physiological conditions, the formation of the open peptides is clearly favored by the acidic gastric environment. This means that the here reported detoxifying effect should be primarily elicited by dietary CHP. This suggests that dietary CHP should play a more effective role when compared to endogenous CHP (or parenterally administered) since the scavenging effects of the open dipeptides should combine with the intrinsic activities exerted by the absorbed CHP. Notably, the open CHP analogues should retain the capacity to promote the zinc absorption since the metal chelation is mostly ascribable to the stereo-electronic properties of the imidazole ring [48].

More generally, the here described results indicate that the reported antioxidant activity against lipid peroxidation can be also ascribed to the scavenging activity of Pro-His [49]. Even more indirectly, these results can also rationalize the neuroprotective effects of TRH and its analogues [50]. Stated differently, the marked scavenging activity exhibited by Pro-His indicates that this peptide can be endowed with all the beneficial effects already documented for l-carnosine [51]. Specifically, the reported anti-inflammatory and neuroprotective effects of CHP can also be ascribable to the detoxifying role of its open derivatives.

From a mechanistic standpoint, the marked reactivity of Pro-His emphasizes that a primary amine (as seen in l-carnosine) is not mandatory and the protonated iminium intermediate resulting from secondary amines enhances the overall scavenging reactivity. This finding remarkably extends the possibility to modify the parent l-carnosine structure by designing compounds bearing variously decorated secondary amines. Finally, the study emphasizes that proline itself and proline containing peptides at their N-terminus might have organocatalytic roles in vivo [52].

## Data Availability

All relevant data are reported in the paper.

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
