# Peer review of "Cyclo(His-Pro) Exerts Protective Carbonyl Quenching Effects through Its Open Histidine Containing Dipeptides"

_nutrients, 2022, doi:10.3390/nu14091775_

Round 1

Reviewer 1 Report

The manuscript need to important correction to be acceptable for publication. Moreover the manuscript include only chemical parameters. No in vivo or in vitro data concerning nutritional consequences are included in the study.

The manuscript need English editing. The meaning of several sentences in the introduction need to be improved.  The numbering of tables and figures needs to be reviewed. Table 2 is before table 1. The citation of figures and tables in the text does not correspond to figures or tables (figure 2A do not exist). 

The study do not include any nutritional data. As indicate by the authors this study investigated the quenching reactivity of TRH, CHP and its open dipeptides towards α,b unsaturated carbonyls. The authors do not have any data to support that CHP-containing food improving insulin sensitivity, body weight control and antioxidant effects. The conclusion need to be rewritten.

Author Response

The manuscript need to important correction to be acceptable for publication. Moreover the manuscript include only chemical parameters. No in vivo or in vitro data concerning nutritional consequences are included in the study.

The manuscript need English editing. The meaning of several sentences in the introduction need to be improved. The numbering of tables and figures needs to be reviewed. Table 2 is before table 1. The citation of figures and tables in the text does not correspond to figures or tables (figure 2A do not exist). 

The manuscript was carefully checked both for the English editing and for numbering of Tables and Figures. The Introduction was improved

The study do not include any nutritional data. As indicate by the authors this study investigated the quenching reactivity of TRH, CHP and its open dipeptides towards α,b unsaturated carbonyls. The authors do not have any data to support that CHP-containing food improving insulin sensitivity, body weight control and antioxidant effects. The conclusion need to be rewritten.

The beneficial effects of CHP-containing food in improving insulin sensitivity, body weight control and antioxidant effects was already reported by three independent studies. Here we describe a mechanism which can rationalize the antioxidant effect of CHP through its open derivatives. The conclusion was almost completely rewritten to better emphasize this point

Reviewer 2 Report

This study investigated the quenching effects of TRH,  cyclo(His-Pro) and its open derivatives towards 4-hydroxy-2-nonenal. Pro-His demonstrated activity and is more reactive than L-carnosine, possibly due to the greater electrophilicity of the involved iminium intermediate.

In my opinion, this study was performed well, with necessary controls. Results are sound.
Minor comments to address: Discussion section is very rudimentary. I would advise extended it with more literature references. E. g. can you elaborate more on this sentence "...On the other hand, these result can rationalize the beneficial effects exerted by the CHP-containing food such as 3improving insulin sensitivity, body weight control and antioxidant effects."- provide references and more insight in quoted sources.

"Finally, the study suggests that proline containing peptides might have multifaceted organocatalytic roles in vivo." Can you provide reference for this statement? Example? appearance of natural CHP in some human diseases or conditions?

Author Response

This study investigated the quenching effects of TRH,  cyclo(His-Pro) and its open derivatives towards 4-hydroxy-2-nonenal. Pro-His demonstrated activity and is more reactive than L-carnosine, possibly due to the greater electrophilicity of the involved iminium intermediate.

In my opinion, this study was performed well, with necessary controls. Results are sound.
Minor comments to address: Discussion section is very rudimentary. I would advise extended it with more literature references. E. g. can you elaborate more on this sentence "...On the other hand, these result can rationalize the beneficial effects exerted by the CHP-containing food such as 3improving insulin sensitivity, body weight control and antioxidant effects."- provide references and more insight in quoted sources.

The discussion was almost completely rewritten

"Finally, the study suggests that proline containing peptides might have multifaceted organocatalytic roles in vivo." Can you provide reference for this statement? Example? appearance of natural CHP in some human diseases or conditions?

References concerning the possible catalytic roles in vivo of proline were included

Reviewer 3 Report

This is an interesting article where the chemical finding could be extrapolated to nutrition and cell metabolism. The chemical part is carefully carried out and the results are sound. However, when the effects of the molecules are assayed toward the hCN1 the results are not clear and need further explanation and probably new experiments.

There is a sentence in the results part that is confusing: “Finally, the hCN1 inhibition by CHP was evaluated by 315 monitoring the residual hCN1 activity when incubating the enzyme with L-carnosine and CHP. The experiment revealed a very poor inhibition since the enzyme retained the 85% of its activity towards L-carnosine”.

1.- Are these data experimental or deduced from the docking experiment? If they are experimental, the protocols used should be added in the materials and methods section.

2.- Furthermore, It would be necessary to carry out the real experiment using either hCN1 enzyme or at least serum, and Dixon plots of the inhibition toward carnosine hydrolysis be presented. Also, it would be advisable to incorporate a fluorimetric method of determination of the hCN1 activity in the measurements.

In addition, the discussion section is too short, and more than a discussion is just a conclusion. I would advise to either fuse to create results and discussion part or expand the discussion including more information regarding the biological significance of the manuscript.

Author Response

This is an interesting article where the chemical finding could be extrapolated to nutrition and cell metabolism. The chemical part is carefully carried out and the results are sound. However, when the effects of the molecules are assayed toward the hCN1 the results are not clear and need further explanation and probably new experiments.

There is a sentence in the results part that is confusing: “Finally, the hCN1 inhibition by CHP was evaluated by 315 monitoring the residual hCN1 activity when incubating the enzyme with L-carnosine and CHP. The experiment revealed a very poor inhibition since the enzyme retained the 85% of its activity towards L-carnosine”.

1.- Are these data experimental or deduced from the docking experiment? If they are experimental, the protocols used should be added in the materials and methods section.

Thank you for the comment. The sentence has been changed to specify the protocol used for the measurement of hCN1 inhibition as follows: “Finally, CHP induced no statistically significant inhibition of hCN1 activity as measured by means of a validated method (10.1016/j.jpba.2020.113440). In fact, serum samples incubated with equimolar amounts of carnosine and CHP retained 85±5% of carnosine hydrolytic activity. The decrease was not statistically significant compared to control samples showing an activity of 1.2 nmol * µL-1 * h-1 ±13%.

2.- Furthermore, It would be necessary to carry out the real experiment using either hCN1 enzyme or at least serum, and Dixon plots of the inhibition toward carnosine hydrolysis be presented. Also, it would be advisable to incorporate a fluorimetric method of determination of the hCN1 activity in the measurements.

As reported above, hCN1 inhibition experiments were performed experimentally by measuring the residual amounts of substrate over time with a validated method (10.1016/j.jpba.2020.113440). CHP was not able to induce a statistically significant decrease of serum hydrolytic activity towards carnosine. In the mentioned paper it is detailed the limitations of the fluorimetric assays and the advantages of the method we used instead.Notice that the primary objective of these analyses was to assess the stability in serum of these peptides. The identification of the involved enzymes is not essential here, although we performed some in silico analyses to evaluate the putative involvement of hCN1 for some peptides.

In addition, the discussion section is too short, and more than a discussion is just a conclusion. I would advise to either fuse to create results and discussion part or expand the discussion including more information regarding the biological significance of the manuscript.

The discussion was almost completely rewritten

Round 2

Reviewer 1 Report

The corrections done really improve the quality of the manuscript.

The conclusion is much more adapted to the manuscript but need English editing. 

Reviewer 3 Report

The manuscript has been improved (mainly the discussion) and the results are more clearly presented. Albeit some results are not completely clear to me (for example the no inhibition of carnosine), the results are interesting enough.